# TRAF1 Deficiency in Macrophages Drives Exacerbated Joint Inflammation in Rheumatoid Arthritis

**DOI:** 10.3390/biom14070864

**Published:** 2024-07-19

**Authors:** Ali Mirzaesmaeili, Ali A. Abdul-Sater

**Affiliations:** School of Kinesiology and Health Science, Muscle Health Research Centre, York University, Toronto, ON M3J 1P3, Canada; alimirzaesmaeili@gmail.com

**Keywords:** TRAF1, rheumatoid arthritis, inflammation, macrophages, collagen antibody-induced arthritis

## Abstract

The tumor necrosis factor receptor-associated factor 1 (TRAF1) plays a key role in promoting lymphocyte survival, proliferation, and cytokine production. Recent evidence showed that TRAF1 plays opposing roles in monocytes and macrophages where it controls NF-κB activation and limits pro-inflammatory cytokine production as well as inflammasome-dependent IL-1β secretion. Importantly, TRAF1 polymorphisms have been strongly linked to an increased risk of rheumatoid arthritis (RA). However, whether and how TRAF1 contributes to RA pathogenesis is not fully understood. Moreover, investigating the role of TRAF1 in driving RA pathogenesis is complicated by its multifaceted and opposing roles in various immune cells. In this study, we subjected wildtype (WT) mice to the collagen antibody-induced arthritis (CAIA) model of RA and injected them intra-articularly with WT- or TRAF1-deficient macrophages. We show that mice injected with TRAF1-deficient macrophages exhibited significantly exacerbated joint inflammation, immune cell infiltration, and tissue damage compared to mice injected with WT macrophages. This study may lay the groundwork for novel therapies for RA that target TRAF1 in macrophages.

## 1. Introduction

Nearly 1% of the general population are afflicted with rheumatoid arthritis (RA), a chronic autoimmune disease characterized by articular inflammation as well as joint and tissue damage. While a variety of innate and adaptive immune cells play a role in the pathogenesis of RA, macrophages are central to initiating and driving the disease [1]. These cells act as primary sources of cytokines (e.g., TNF and IL-1β), chemokines, and degradative enzymes, contributing to joint inflammation and ultimately leading to the degradation of cartilage and bone. Importantly, the number of macrophages in synovial tissue is of clinical importance, serving as a reliable marker for assessing disease severity and therapy response [1].

Genetic polymorphisms can influence the incidence and severity of RA. Genome-wide association studies have strongly linked a region on chromosome 9 in the TRAF1/C5 locus tagged by several single-nucleotide polymorphisms (SNPs) in linkage disequilibrium, including rs3761847, with an elevated risk of rheumatoid arthritis [2,3]. Moreover, RA patients harboring these SNPs exhibit higher mortality from sepsis [4]. A recent study has identified rs7034653 in TRAF1 as the functional variant linked to increased susceptibility to RA [5]. TRAF1 is an important signaling adapter protein that plays disparate roles in various immune cells. TRAF1 can promote proliferation and survival of T and B lymphocytes downstream of TNFR signaling (e.g., 4-1BB and CD40) but inhibit inflammatory cytokine production from monocytes and macrophages downstream of innate immune signaling [6,7]. A key study showed that healthy individuals with the disease-associated SNP exhibit decreased TRAF1 expression in their peripheral blood monocytes and T lymphocytes [8]. Consistent with TRAF1’s pro-survival role in lymphocytes, T cells from individuals with the disease-associated SNP produced fewer cytokines following activation. On the other hand, monocytes from those same individuals produced higher levels of pro-inflammatory cytokines (TNF, IL-6) following lipopolysaccharide (LPS) stimulation compared to those from the lower risk group. Importantly, this study showed that the increase in pro-inflammatory cytokines from monocytes due to reduced TRAF1 levels outweighed the reduction in cytokines produced from T cells, leading to an overall pro-inflammatory state. This study went on to demonstrate that TRAF1 interferes with a critical step in toll-like receptor (TLR)-induced NF-κB activation. TRAF1 directly binds and lowers the formation of the linear ubiquitin chain assembly complex (LUBAC), an important ubiquitin ligase that catalyzes the linear (M1-linked) poly-ubiquitination of NEMO (a.k.a. IKK-**γ**), the regulatory subunit of the IKK complex. Using TRAF1 knockout mice, this study showed that macrophages lacking TRAF1 displayed increased NF-κB activation and cytokine production after TLR stimulation, and that TRAF1-deficient mice were more susceptible to mortality from sepsis [8].

Crucially, this study did not investigate whether the lower levels of TRAF1 in macrophages directly contribute to rheumatoid arthritis pathogenesis. In fact, using TRAF1 knockout mice to determine if and how it affects a complex autoimmune disease, like rheumatoid arthritis, will be complicated by the opposing role that TRAF1 plays in monocytes/macrophages versus T/B lymphocytes. Consistent with the multifaceted role of TRAF1, a study by Cheng et al. employed the KRN/I-A(g7) (KxB/N) model of RA and demonstrated that TRAF1 knockout mice developed equally progressive and severe arthritis to that observed in wildtype mice [9]. Interestingly, TRAF1 knockout mice had markedly lower levels of anti-GPI antibodies, which is consistent with a positive role for TRAF1 in promoting lymphocyte activation [9]. Therefore, a model that studies the role of TRAF1 in macrophages without interference from lymphocytes is needed to properly evaluate how TRAF1 contributes to disease pathogenesis.

In this report, we sought to determine for the first time whether TRAF1’s ability to limit inflammatory cytokine production in macrophages contributes to its role in RA pathogenesis. We show that intra-articular injection of bone marrow-derived macrophages prepared from TRAF1 knockout mice exacerbated joint inflammation, bone erosion and inflammatory cell recruitment in the collagen antibody-induced arthritis (CAIA) model of RA. To our knowledge, this is the first report that implicates TRAF1 in modulating disease in a model of RA.

## 2. Materials and Methods

### 2.1. Cell Culture and Reagents

Primary bone marrow-derived macrophages (BMDMs) were prepared from C57/Bl6 wildtype (WT) and TRAF1^−/−^ mice and cultured, as described previously [10,11]. Briefly, the femur and tibia of mice were flushed and cultured in RPMI 1640 (Sigma, St. Louis, MO, USA) supplemented with 10% fetal bovine serum (Wisent, Saint-Jean-Baptiste, QC, Canada), 2-Mercapthoethanol (Gibco, Grand Island, NY, USA), L-Glutamine (Sigma), Pyruvate (Sigma), penicillin (Sigma), streptomycin (Sigma), non-essential amino acids (Gibco), and 25% L929-conditioned media.

### 2.2. Histological Examinations

Knee joints were dissected and fixed in 10% formalin neutral buffer solution (Thermo Fisher Scientific, Waltham, MA, USA) overnight and decalcified with Rapid Decalcifier Solution (RDO; Apex Engineering Products, Aurora, IL, USA) for 2 h, washed, and then transferred to 70% ethanol and embedded in paraffin (Thermo Fisher Scientific). Paraffin blocks were then sectioned (5 μm) and deparaffinized in xylene (Thermo Fisher Scientific) then stained with hematoxylin (Sigma) and eosin (Thermo Fisher Scientific), as previously described [10]. The slides were scanned (EVOS FL-Auto, Thermo Fisher Scientific), and quantification of inflammatory cells was performed in a blinded fashion.

### 2.3. Mice and CAIA Model

We chose to employ the collagen antibody-induced arthritis (CAIA) model of RA because it mimics the inflammatory phase of human rheumatoid arthritis. This is achieved because the injected monoclonal antibodies induce the formation of immune complexes, which further activate monocytes and trigger the release of pro-inflammatory cytokines. These cytokines then recruit macrophages and neutrophils to the joint, replicating the inflammatory response seen in human RA. To this end, 7–8 weeks old C57/Bl6 wildtype male mice (18–20 g; Charles River) were injected interperitoneally (i.p.) with either 2.5 or 5 mg monoclonal antibody cocktail against type II collagen (Arthrogen-CIA^®^ Arthritogenic Monoclonal Antibody, Chondrex, Woodinville, WA, USA) on day 0. This was followed by intra-articular (i.a.) injection with PBS in the left knee or with 2.5 × 10^5^ BMDMs from WT mice or TRAF1^−/−^ mice in 10 uL PBS in the right knee (N = 3 mice per group). Some mice received i.a. injection of WT BMDMs in the left knee and TRAF1^−/−^ BMDMs in the right knee (N = 3 mice per group). Among other things, injections were performed using 30-gauge Hamilton syringes (Hamilton Company, Reno, NV, USA) into contralateral sides [10]. The animals’ arthritis score, body weight, and grimace scale were evaluated daily. Knee and ankle measurements were performed on days 0, 3 and 6 using a 3-Mode Digital Fractional Caliper (Husky, Pacific, MO, USA).

### 2.4. Statistical Analyses

All statistical analyses were performed using GraphPad software (Prism version 9) using a 2-way analysis of variance (ANOVA) for comparison of multiple groups with multiple comparisons with Sidak correction. *p*-values are indicated in the figure legends.

## 3. Results

### 3.1. Macrophages Injected Intra-Articularly into the Knee Joint Become Resident in the Synovial Tissue

Since TRAF1 plays opposing roles in innate signaling in inflammatory cells compared to TNFR signaling in adaptive lymphocytes, we wanted to design a model where we can test its effects in macrophages without the contribution from other immune cells. Therefore, we prepared macrophages from the bone marrow of wildtype (WT) mice and injected them intra-articularly into the knee joints of WT C57/Bl6 mice. We first examined whether the injected macrophages reside in the synovial space and for how long. Indeed, we observed that CFSE-labeled macrophages remained resident in the synovial tissues for at least 72 h post injection (Figure 1).

### 3.2. Intra-Articular Injection of TRAF1-Deficient Macrophages Exacerbates Joint Inflammation in CAIA

Next, to determine whether TRAF1 plays a role in the contribution of macrophages to RA pathogenesis, we injected WT and TRAF1 knockout (TRAF1^−/−^) macrophages into the knee joint intra-articular space of WT C57/Bl6 mice. Intra-articular injection of PBS served as the vehicle control. This was followed by intraperitoneal (i.p.) injection of a low dose (2.5 mg) or a high dose (5 mg) of Arthrogen-CIA^®^ arthritogenic monoclonal antibody cocktail to trigger the collagen antibody-induced arthritis (CAIA) model of RA. Intra-articular injection of PBS in one knee joint did not induce additional knee joint swelling compared to knees with no intra-articular injections (Figure 2A). Contrastingly, mice injected with WT macrophages exhibited more swelling in their knee joints than those injected with PBS in the contralateral side (*p* < 0.05 at day 6; Figure 2B). Remarkably, intra-articular injection of TRAF1^−/−^ macrophages led to a significant increase in knee joint inflammation compared to the contralateral side injected with PBS (*p* < 0.0001 at days 5 and 6; Figure 2C) and WT macrophages (*p* < 0.0001 at days 5 and 6; Figure 2D). The increase in joint inflammation driven by the macrophages did not extend to nearby ankle joints that received no injections (Figure 2E,F).

### 3.3. Increased Cellular Infiltration, Angiogenesis, and Bone Erosion in Knee Joints of CAIA Mice Injected with TRAF1-Deficient Macrophages

We then examined whether the increase in knee joint swelling following intra-articular injection of macrophages was accompanied with an increase in cellular infiltration and tissue damage. Indeed, we observed that there was a significant increase in inflammatory cell infiltrates in the synovia of knee joints injected with TRAF1^−/−^ macrophages compared to those injected with WT macrophages and PBS (Figure 3A). This was accompanied by an increase in angiogenesis and bone erosion, indicating that the presence of TRAF1^−/−^ macrophages exacerbated tissue damage in the knee joints when compared to WT macrophages and PBS controls (*p* < 0.05; Figure 3B).

## 4. Discussion

Numerous studies have established a connection between single-nucleotide polymorphisms (SNPs) in the *TRAF1* gene and an increased risk of developing rheumatoid arthritis [2,3,4,12,13]. TRAF1 plays multifaceted roles in various immune cells and downstream of distinct immune signaling pathways [6,7]. In a more recent study, we employed a monosodium urate (MSU) crystal-induced model of gout in mice and reported that TRAF1-deficient mice exhibited a significant increase in joint swelling and inflammatory cell infiltration into the synovium compared to their wildtype littermates [10]. Mechanistically, TRAF1 lowered IL-1β secretion and inhibited the NLRP3 inflammasome assembly in macrophages by limiting the linear ubiquitination of the adapter protein, ASC. Yet, to our knowledge, no studies have demonstrated a direct involvement of TRAF1 in driving the pathogenesis of RA. In this study, we showed that macrophages deficient in TRAF1 can significantly exacerbate joint inflammation and tissue damage when injected into the intra-articular space of knee joints of mice subjected to the collagen antibody-induced arthritis (CAIA) model of RA.

A multitude of immune cells play an important role in the pathogenesis of complex autoimmune diseases, like RA. However, macrophages and T lymphocytes are recognized as key drivers of the disease [1,14]. Importantly, macrophages and their products are implicated in synovial angiogenesis, a process crucial to the development of RA [15]. Investigating the role of TRAF1 in RA is complicated by TRAF1’s opposing roles in immune signaling, where it promotes T cell activation and proliferation downstream of 4-1BB, while inhibiting inflammation in monocytes and macrophages downstream of TLR signaling [7]. Indeed, in one study, TRAF1 knockout mice were equally susceptible as wildtype mice to a spontaneous model of RA [9]. The researchers noted that TRAF1-deficient mice had lower anti-GPI antibody titers and that this was due to the decrease in activation of B and T cells. This is consistent with the positive role that TRAF1 plays in lymphocyte activation. However, this study failed to explain the comparable disease between WT and TRAF1 knockout mice. We believe that it was likely increased inflammation from macrophages in knockout mice that compensated for the decrease in reduced antibody titers. Therefore, to isolate the effects of TRAF1 in macrophages in RA, we deliberately injected WT or TRAF1 knockout macrophages into the knee joint of WT mice followed by induction of the CAIA model of RA. Remarkably, TRAF1 knockout macrophages increased knee joint but not ankle joint inflammation compared to WT macrophages. Knees injected with TRAF1 knockout macrophages also exhibited exacerbated tissue damage as indicated by increased bone erosion and angiogenesis. The limitations of this study include the injection of bone marrow-derived macrophages into the knee joints. Investigating the cell-specific role of TRAF1 in RA is hindered by the lack of conditional TRAF1 knockout mice, and future studies should focus on generating these mice to confirm these observations in synovial macrophages.

While great progress has been made in understanding RA pathogenesis, a cure remains elusive. Further research into the molecular mechanisms driving RA inflammation is vital. This study points to TRAF1 in macrophages as a potential therapeutic target for reducing inflammation and joint damage in RA.

## Figures and Tables

**Figure 1 biomolecules-14-00864-f001:**
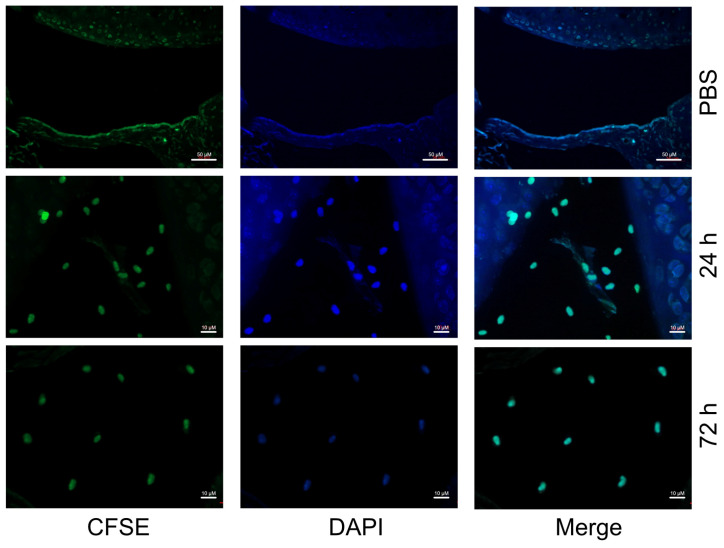
BMDMs persist in the synovium after intra-articular injection. An amount of 10^5^ bone marrow-derived macrophages (BMDMs) were prepared from wildtype (WT) C57/Bl6 mice and labeled with 1μM CFSE (green) immediately prior to intra-articular injection into the knee joints of WT mice. PBS was injected as a negative control. Synovial tissues from knee joints were dissected 24 h or 72 h post injection and stained with the nuclear stain DAPI (blue) and were visualized by confocal microscopy. Images are representative of 7 mice per group.

**Figure 2 biomolecules-14-00864-f002:**
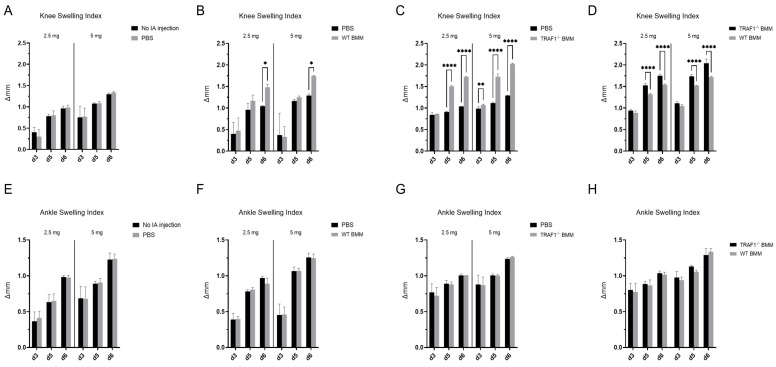
Increased knee joint swelling following injection of TRAF1-deficient macrophages in CAIA mice. WT mice were injected intra-articularly with PBS in one knee joint (**A**,**E**), with 2.5 × 10^5^ WT BMDMs in one knee joint and, as a control, PBS in the contralateral knee joint (**B**,**F**), with 2.5 × 10^5^ TRAF1^−/−^ BMDMs in one knee joint and, as a control, PBS in the contralateral knee joint (**C**,**G**), or with 2.5 × 10^5^ WT BMDMs in one knee joint and an equal number of TRAF1^−/−^ BMDMs in the contralateral knee joint (**D**,**H**). The mice were then injected intraperitoneally with either 2.5 mg or 5 mg of the Arthrogen-CIA^®^ arthritogenic monoclonal antibody cocktail to induce CAIA. Knee (**A**–**E**) and ankle (**F**–**H**) joint thickness was measured before the injections and at 3, 5 and 6 days after injection using a digital caliper. Data were reported as the change in thickness (Δ mm). n = 3 per group. Data in all graphs were compared using a two-way ANOVA with multiple comparisons with Sidak correction. * *p* < 0.05, ** *p* < 0.01, **** *p* < 0.0001.

**Figure 3 biomolecules-14-00864-f003:**
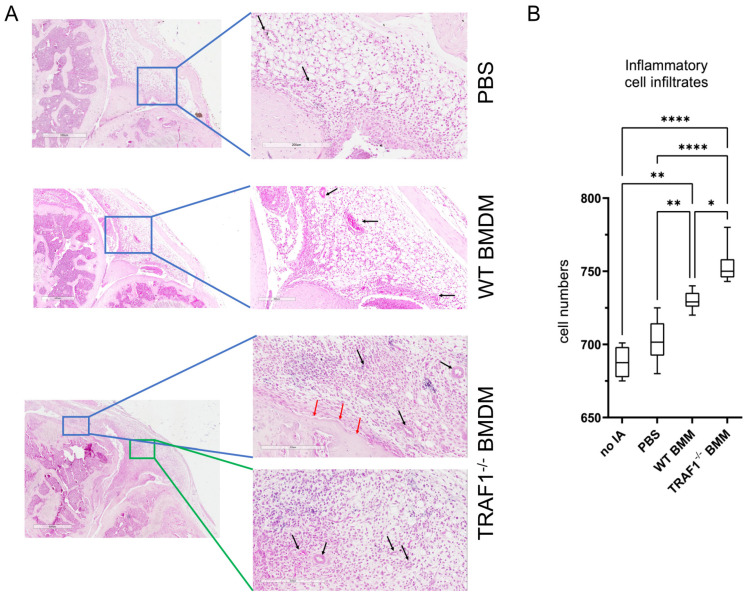
Injection of TRAF1-deficient macrophages exacerbates inflammatory cell infiltration and bone damage in CAIA mice. WT mice were injected intra-articularly with 2.5 × 10^5^ WT or TRAF1^−/−^ BMDMs in one knee joint and, as a control, PBS in the contralateral knee joint followed by intraperitoneal injection with 5 mg of the Arthrogen-CIA^®^ arthritogenic monoclonal antibody cocktail to induce CAIA. (**A**) Histopathological evaluation of CAIA mice showing representative H&E staining images for synovial tissue of the knee joint sections injected with PBS, WT, or TRAF1^−/−^ BMDMs. The results shown are representative images from three mice per group. Black arrows indicate angiogenesis; red arrow indicate bone erosion (**B**) Quantification of the average inflammatory infiltrates per section, as in (**A**). Statistical analysis was performed using two-way ANOVA with multiple comparisons with Sidak correction. * *p* < 0.05, ** *p* < 0.01, **** *p* < 0.0001.

## Data Availability

Data is contained within the article.

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
