# Peer review of "TRAF1 Deficiency in Macrophages Drives Exacerbated Joint Inflammation in Rheumatoid Arthritis"

_biomolecules, 2024, doi:10.3390/biom14070864_

Round 1

Reviewer 1 Report

Comments and Suggestions for Authors

In this study the Mirzaesmaeili et al., performed an animal study to define whether TRAF1 contributes to rheumatoid arthritis (RA) pathogenesis via using an arthritis (CAIA) model of mice, induced by collagen antibody. They found that when the TRAF1 deficient macrophages were injected into knee joints of the mice model, it significantly exacerbated joint inflammatory response, which confirmed that TRAF1 protein in expressed in macrophages are able to limit inflammatory response. The methods, such as immunohistochemistry, are correctly used. The novelty is that these studies were performed in vivo.  Though the data support the conclusions. However, If  the photos of the keen joints, and expression levels of proinflammatory cytokines in the joint keen were provided, they would further supporte the findings.

Line 62 please delete the repeated word, “ using”.

Author Response

Here we address the reviewers’ comments (in italics), with a response in plain text.

  1. Though the data support the conclusions. However, If the photos of the keen joints, and expression levels of proinflammatory cytokines in the joint keen were provided, they would further supporte the findings.

We thank the reviewer for this suggestion. We have photos of the knee joints (please see attached file to this response); however, we made the determination that it is difficult to assess the severity of the knee joint swelling from these photos so we chose not to include them in the figure. Here is the figure showing a representative knee joint from each group. If the reviewer insists on including them in the manuscript, we can add them as a supplementary figure.

Unfortunately, we did not extract RNA from the knee joints that would have allowed us to measure inflammatory gene expression. This was in part due to unreliability of inflammatory gene expression as a measure of inflammation severity in the CAIA model at this time point. 

  1. Line 62 please delete the repeated word, “ using”.

The word has been deleted

Reviewer 2 Report

Comments and Suggestions for Authors

I read the manuscript entitled “TRAF1 Deficiency in Macrophages Drives Exacerbated Joint Inflammation in Rheumatoid Arthritis” with great interest. It outlines the role of tumor necrosis factor receptor-associated factor 1 in joint inflammation, bone erosion, and inflammatory cell recruitment in the collagen antibody-induced arthritis (CAIA) model of rheumatoid arthritis. I have some minor concerns that could enhance the readability of this study for all prospective readers:

1.     Some lines are overly long; splitting them could improve the flow. Specifically, lines 15-18 and 40-43.

2.     I strongly recommend that the authors revise the introduction section; the current details could be incorporated into the discussion section.

3.     There is a spelling error in line 63 ("Knockout").

4.     In the methods section, providing additional context or references regarding the rationale for using this model/treatments would be beneficial.

5.     Lines 108 and 109 do not fit properly. Consider rephrasing the sentence to clarify the intended meaning of 'N = 3 mice per group.' Alternatively, you may place 'N = 3' in brackets at a relevant location within the text.

6.     In the results section, although the author mentions that the p-values are given in the figure legends, it might be helpful to include p-values within the text for clarity.

7.     The discussion section seems brief.

8.     The statement on lines 191-192 mentions that "no studies have demonstrated a direct involvement of TRAF1 in driving the pathogenesis of RA." However, later in the text, all statements demonstrate the opposite trend. This appears contradictory. Clarifying this point would help readers understand the significance of the research findings.

9.     A more explicit discussion of the limitations of the current study and potential avenues for future research is needed to enhance clarity further.

Author Response

Here we address the reviewers’ comments (in italics), with a response in plain text.

  1. Some lines are overly long; splitting them could improve the flow. Specifically, lines 15-18 and 40-43. 

We thank the reviewer for their suggestions on improving the text. We split these sentences as suggested by the reviewer. 

  1. I strongly recommend that the authors revise the introduction section; the current details could be incorporated into the discussion section.

We revised the section to remove some details that would fit better in the discussion.

  1. There is a spelling error in line 63 ("Knockout").

This has been corrected

  1. In the methods section, providing additional context or references regarding the rationale for using this model/treatments would be beneficial.

We added a short paragraph in the methods explaining the rationale/advantages of use the CAIA model, as suggested.

  1. Lines 108 and 109 do not fit properly. Consider rephrasing the sentence to clarify the intended meaning of 'N = 3 mice per group.' Alternatively, you may place 'N = 3' in brackets at a relevant location within the text.

We thank the reviewer for spotting this error. This has not been clarified. 

  1. In the results section, although the author mentions that the p-values are given in the figure legends, it might be helpful to include p-values within the text for clarity.

We added the p values in the results, as suggested

  1. The discussion section seems brief.

The discussion has been slightly expanded as we moved some text from the introduction as suggested in point # 2. However, additional expansion of the discussion section was not possible due to the stringent character limit for this type of “Brief report”.

  1. The statement on lines 191-192 mentions that "no studies have demonstrated a direct involvement of TRAF1 in driving the pathogenesis of RA." However, later in the text, all statements demonstrate the opposite trend. This appears contradictory. Clarifying this point would help readers understand the significance of the research findings. 

The statement we are making is that no studies have show a direct role for TRAF1 in RA, which is true. All previous studies, as explained in the intro and discussion have shown an association or correlation between TRAF1 and RA, but no studies have show a direct role in pathogenesis.

  1. A more explicit discussion of the limitations of the current study and potential avenues for future research is needed to enhance clarity further. 

The limitations have now been discussed near the end of the discussion section.